# Genotype–Phenotype Correlation of GNAS Gene: Review and Disease Management of a Hotspot Mutation

**DOI:** 10.3390/ijms252010913

**Published:** 2024-10-10

**Authors:** Lorenzo Cipriano, Rosario Ferrigno, Immacolata Andolfo, Roberta Russo, Daniela Cioffi, Maria Cristina Savanelli, Valeria Pellino, Antonella Klain, Achille Iolascon, Carmelo Piscopo

**Affiliations:** 1Department of Molecular Medicine and Medical Biotechnology, University Federico II, 80131 Naples, Italy; lorenzo.cipriano@unina.it (L.C.); immacolata.andolfo@unina.it (I.A.); roberta.russo@unina.it (R.R.); achille.iolascon@unina.it (A.I.); 2Endocrinology and Auxology Unit, Pediatrics Speciality Department, A.O.R.N. Santobono-Pausilipon, 80131 Naples, Italy; r.ferrigno@santobonopausilipon.it (R.F.); d.cioffi@santobonopausilipon.it (D.C.); mc.savanelli@santobonopausilipon.it (M.C.S.); v.pellino@santobonopausilipon.it (V.P.); a.klain@santobonopausilipon.it (A.K.); 3Medical and Laboratory Genetics Unit, A.O.R.N. “Antonio Cardarelli”, 80131 Naples, Italy

**Keywords:** GNAS, PHP1a, c.565_568delGACT, intellectual disability, heterotopic ossifications

## Abstract

Defects of the *GNAS* gene have been mainly associated with pseudohypoparathyroidism Ia. To date, pathogenic missense, frameshift, non-sense and splicing variants have been described in all the 13 exons of the *GNAS* gene. Of them, a specific mutation, namely the 4 bp deletion c.565_568delGACT, is currently considered a mutation hotspot. Recent articles performed genotype–phenotype correlations in patients with *GNAS*-related pseudohypoparathyroidism Ia (PHP1a) but a specific focus on this hotspot is still lacking. We reported two cases, from our department, of PHP1a associated with c.565_568delGACT deletion and performed a literature review of all the previously reported cases of the 4 bp deletion hotspot. We found a higher prevalence of brachydactyly, round face, intellectual disability and subcutaneous/heterotopic ossifications in patients with the c.565_568delGACT as compared to the other variants in the *GNAS* gene. The present study highlights the different prevalence of some clinical features in patients with the c.565_568delGACT variant in the *GNAS* gene, suggesting the possibility of a personalized diagnostic follow-up and surveillance for these patients.

## 1. Introduction

*GNAS* gene is located on the long arm of the chromosome 20, consists of 13 exons, and encodes several gene products, including the alpha-subunit of the stimulatory guanine nucleotide-binding protein (Gsα), the extra-large Gsα (XLαs), and the neuroendocrine secretory protein 55 (NESP55) [1]. It is an imprinted gene and the expression of the paternal allele is reduced/silenced in some tissues, such as the pituitary and thyroid glands, gonads, and proximal renal tubules [2]. Therefore, most *GNAS*-related products are mainly derived from the maternal allele and this explains why maternal-related mutations are associated with an important reduction in *GNAS* products in some tissues and, consequently, with specific diseases [3].

Of all the genetic products of the *GNAS* gene, Gsα is commonly considered the most clinically relevant and its dysfunctions are the principal cause of the *GNAS*-related disorders [4]. This is mainly due to the important role that Gsα plays in signal transduction pathways in numerous human glands. In fact, Gsα is required for membrane signal transduction of several hormones such as the parathyroid hormone (PTH), the thyroid stimulating hormone (TSH), gonadotropins, and the growth hormone-releasing hormone (GHRH); a Gsα dysfunction can determine resistance to these hormones of variable severity [5].

Defects of the *GNAS* gene have been associated with different phenotypes such as pseudohypoparathyroidism Ia, Ib, and Ic (PHP-Ia, -Ib, -Ic), pseudopseudohypoparathyroidism (PPHP), progressive osseous heteroplasia (POH), and osteoma cutis (OC) [3]. Of them, PHP is the most frequent and is currently divided into two main subtypes (i.e., type 1 and type 2). The first is defined as the failure to increase the excretion in urine of cAMP and phosphate following PTH administration whereas in PHP type 2, the cAMP response is in the normal range, with a reduced urinary phosphate excretion in response to PTH [6].

PHP type 1 is divided into subtypes a, b, and c. This distinction is firstly based on the presence (subtype 1a and 1c) or absence (type 1b) of Albright hereditary osteodystrophy (AHO). PHP1a and PHP1c are distinguished on the basis of the Gsα activity in erythrocytes (reduced in PHP1a, normal in PHP1c). PPHP shares several clinical features with PHP-Ia, with various manifestations of the AHO phenotype, but without the hormone resistance or obesity. So, the presence or absence of hormonal resistance differentiates PHP1 from PPHP. POH is a dermal ossification that starts in infancy and is followed by progressive increase and extensive bone formation in deep muscle and fascia. OC consists of extra-skeletal ossification restricted to the dermis and subcutaneous tissues. In 2016, the EuroPHP network proposed a new classification with a more uniform terminology [6], summarizing all these disorders under the term “inactivating PTH/PTHrP signaling disorder” (iPPSD). For the diagnosis of iPPSD, a minimum of one of the major criteria (PTH resistance, ectopic ossification, brachydactyly type E) is mandatory. For only brachydactyly as a major criteria, due to the non-infrequent description in other disorders, at least two other minor criteria (THS resistance, other hormonal resistances, motor and cognitive restriction or impairment, intrauterine and postnatal growth restriction, obesity/overweight, flat nasal bridge and/or maxillary hypoplasia, and/or round face) are required [6].

According to this more recent classification, the clinical presentations determined by defects in the *GNAS* expression and in the Gsα function, are classified as iPPSD2, which includes the previously called PHP type 1a and 1c, PPHP, POH, and OC [6]. Epigenetic defects at the *GNAS* locus determine the iPPSD3 (the previous PHP1b).

The first description of PHP1a associated with a *GNAS* mutation dates back to 1990 [7]. To date, pathogenic missense, frameshift, non-sense and splicing variants have been described in all the 13 exons of the *GNAS* gene. The most frequent associated phenotype of the *GNAS* spectrum disorders is the PHP1a and the highest frequency of pathogenic variants was mainly found in the exons 1 and 7 [8]. The high occurrence of pathogenic variants in exon 7 is currently considered to be a consequence of the high frequency of a specific deletion in this exon. The 4 bp deletion (c.565_568delGACT) in exon 7 has been reported as a hotspot in several multicenter studies [9]. Although Jiang et al., in a recent article, performed an extensive genotype–phenotype correlation in patients with PHP1a [8], describing the prevalence of the most relevant signs and symptoms according to the mutation type, a specific focus on this mutation hotspot is still lacking. Therefore, the aim of the present study is to provide a genotype–phenotype correlation by reporting the prevalence of the most relevant clinical features due to the 4 bp deletion (c.565_568delGACT) in exon 7. To do this, we performed a literature review of all the previous clinical descriptions associated with the c.565_568delGACT variant in the *GNAS* gene. The review was enriched with a description of two unrelated patients, referring to our hospital, affected by PHP1a associated with the c.565_568delGACT pathogenic variant in the *GNAS* gene.

## 2. Materials and Methods

### 2.1. Literature Review

We searched in the PubMed database for all the reported cases of the 4 bp deletion hotspot.

The research was conducted starting from 1 January 1990 (the year of the first description of a *GNAS* variant) to 15 December 2023 and limited to the English language. The search strategy included the Boolean operators on the following terms “*GNAS*”, “*GNAS1*”, “pseudohypoparathyroidism”, “PHP1”, “AHO”, “pseudopseudohypoparathyroidism”, “PPHP”, and “osteoma cutis”. The reference lists of the main studies (including meta-analyses and reviews) were also screened. Bioinformatic tools and databases (e.g., Varsome, Franklin, Gnomad, etc.) were screened to extract adjunctive articles not identified with the previous research method.

The inclusion criteria were (1) studies performed on human subjects only; (2) cohort, case control, and cross-sectional studies, case reports, case series, and the literature, as well as systematic reviews and meta-analyses, including *GNAS* mutations; (3) studies that provided extensive description of the phenotype (e.g., clinical presentation, laboratory data, AHO features, etc.); and (4) studies published in the English language only.

Exclusion criteria were (1) studies that provided insufficient information about the clinical characteristics; and (2) studies including mutations of the c.565_568delGACT variant.

Two reviewers (LC and CP) assessed the eligibility of the identified studies. The review process was articulated in two main phases. The first step consisted of a screening of titles and abstracts obtained through PubMed database searching, followed by a full-text review of selected articles. The two reviewers performed both the steps independently. A third review author was consulted in case of disagreement, dealing with the final decision. Adjunctive articles were added on the basis of cross referencing.

### 2.2. Genetic Testing

DNA samples were obtained from each subject after signed informed consent, and according to the Declaration of Helsinki. Genomic DNA was prepared and clinical exome analysis was conducted using a commercially available panel of over 5000 genes associated with hereditary diseases (SureSelect custom Constitutional Panel 17 Mb, Agilent Technologies), as previously described [10,11]. Variant pathogenicity was assessed following the ACMG/AMP guidelines using a scaled point system [12,13]. Scores ranged from 0 to 5 for variants of unknown significance (VUS), and between 6 and 9 for likely pathogenic (LP) variants. Variants scoring 10 or higher were classified as pathogenic (P). Pathogenic variants related to the phenotype were confirmed by Sanger sequencing.

## 3. Results

### 3.1. Case Description

#### 3.1.1. Patient 1

An eight-month-old male infant was admitted to our department due to neonatal obesity.

He was delivered by caesarean section at 36 weeks and five days of pregnancy, with a birth weight, length, and head circumference of 2760 gr (−0.38 SD), 47 cm (−0.67 SD), and 33.5 cm (−0.12 SD), respectively. During pregnancy, due to abortion menace in the first trimester, his mother was treated with progesterone and acetylsalicylic acid, and in the second trimester she was infected by COVID-19, but did not require any specific treatment, as the infection was only mildly symptomatic, with asthenia, nasal discharge, fever, and myalgias lasting for less than a week. At birth, the patient experienced hypoglycemia and jaundice, treated with supportive care and phototherapy for about 48 h. After 48 h, he also showed bilious vomiting, requiring intensive care unit admittance and care for about one week. At four months of age, he was again admitted to the intensive care unit due to bronchiolitis, requiring treatment with corticosteroids and respiratory support for about two weeks. After discharge, the patient showed an increased growth rate, with a rapid weight gain (about 5 kg in three months), thus influencing his pediatrician to seek endocrinological evaluation.

At presentation, the patient’s weight, length, and head circumference were 11.7 kg (2.4 SD), 65.6 cm (−1.84 SD), and 46.7 cm (1.44 SD), respectively. At physical examination, he showed round face, short neck, brachydactyly, and a single subcutaneous, calcific lesion in his right anterior thorax; no other pathological evidence was found. An endocrine assessment showed normal calcium, slightly increased TSH and IGF-1, reduced FT4 and serum cortisol levels, and negative anti-thyroid autoantibodies; the PTH levels were normal, although close to the upper limit of normal (ULN). A second assessment confirmed hypothyroidism; therefore, levothyroxine replacement was started at 2.13 mcg/kg/die, while an ACTH stimulation test excluded adrenal insufficiency. After one month of levothyroxine treatment, normal FT4 levels were reached (1.09 ng/dL), although the TSH levels were still slightly high (5.71 mcUI/mL), but a further increase in weight (+700 gr) and a persistent, increased growth rate (5.9 cm gain in one month) were observed.

A four-month follow-up was started. At 13 months of age, the patient’s weight further increased to 14.4 kg (2.72 SD) and the growth rate was still increased, although it was less than it was in previous findings (17.1 cm/year). At physical examination, new subcutaneous, calcific lesions were identified at the upper lip and at the right leg. His mother reported pneumonia, at 12 months, that was treated with corticosteroids and amoxicillin/clavulanic acid for about one–two weeks. An endocrine assessment showed normal calcium, increased TSH, IGF-1, and PTH, and reduced Vitamin D and FT4 levels at the ULN, whereas a cortisol assessment was not performed due to recent corticosteroid treatment. Therefore, levothyroxine treatment was maintained at the same dosage and vitamin D supplementation with alpha-calcidol at 0.02 mcg/kg/die was started. Due to the increase in PTH levels, the occurrence of new subcutaneous, calcific lesions, and the clinical phenotype, pseudohypoparathyroidism, was suspected and the patient was referred for a genetic assessment.

The performed clinical exome revealed the presence of the variant c.565_568delGACT in the *GNAS* gene. The mutation was classified as pathogenic accordingly (ACMG) and has been previously described in several other cases of PHP1a.

#### 3.1.2. Patient 2

A one-month-old male newborn was admitted to our department due to positivity in the newborn screening for congenital hypothyroidism.

He was delivered by caesarean section at 40 weeks and one day after a previously uneventful pregnancy due to oligohydramnios, with a birth weight, length, and head circumference of 3095 gr (−0.89 SD), 50 cm (−0.33 SD), and 35 cm (0.27 SD), respectively. At birth, the patient experienced hypoglycemia, hypocalcemia, and jaundice, treated with supportive care.

At presentation, the patient’s weight, length, and head circumference were 4.9 kg (0.74 SD), 54.7 cm (0.01 SD), and 38.7 cm (0.3 SD), respectively. At the physical examination, no pathological evidence was found. An endocrine assessment showed normal calcium and FT4 and increased TSH levels. Therefore, levothyroxine replacement was started at 5.1 mcg/kg/die. After one month of levothyroxine treatment, normal FT4 levels were maintained (1.09 ng/dL), although the TSH levels were still slightly high (5.71 mcUI/mL), so levothyroxine treatment was maintained at the same dosage. The growth rate (1.7 cm/month) and weight increase (+1220 gr) were consistent with age.

A monthly follow-up was started, showing, during the first year, a regular growth rate (25.7 cm/year), whereas the weight gain started to be higher than expected upon four months of age (+2 kg every 3 months). Over time, round face, short neck, and brachydactyly became evident at the physical examinations. The levothyroxine treatment was gradually reduced to 2.8 mcg/kg/die, according to minimum efficacy dose. Calcium was persistently normal throughout the year.

At the one-year follow-up, the physical examination showed a new-onset, single, subcutaneous, calcific lesion in the upper abdomen, so an endocrine assessment for PTH was performed, showing a marked increase (162 pg/mL). Due to the increase in PTH levels and the clinical phenotype, pseudohypoparathyroidism was suspected and the patient was referred for genetic assessment.

As for the first patient, a clinical exome was conducted and revealed the presence of the maternally inherited variant c.565_568delGACT in the *GNAS* gene.

### 3.2. Literature Review

A total of 2080 articles were found through PubMed database searching (Figure 1). After title and abstract evaluation, 230 articles were considered eligible for extensive full-text screening. Most of the excluded studies reported clinical presentations of PHP1a associated with mutations in the *GNAS* gene different from the c.565_568delGACT. A small percentage of them provided an incomplete clinical description of the case. A total of 22 were included in the current review.

The clinical information of a total of 76 patients from 22 studies were extracted. Two additional individuals (our patients) were added, totaling 78 clinical presentations associated with the 4 bp deletion (c.565_568delGACT) in exon 7. Two patients were excluded because of diagnostic ambiguity (two diagnoses of PPHP/POH by [14]).

Of the remaining 76 patients, 61 subjects received a diagnosis of PHP1a, 11 of POH, three of PPHP and one of AHO (Figure 2a). Most of the studies that described clinical presentations other than PHP1a provided poor clinical information about their patients; so, we focused on PHP1a for our genotype–phenotype correlation study.

The mean age at the onset of one of the signs/symptoms related to PHP1a was 6.10 years, with male predominance (62% of the cases). Hypocalcemia at diagnosis was present in 47% of the cases, hyperphosphatemia in 42%, and an elevated PTH at diagnosis in 64% of the cases. Of the commonly described clinical features of Albright hereditary osteodystrophy, brachydactyly showed the highest prevalence (74%), followed by round face (71%), intellectual disability (69%), and subcutaneous/heterotopic ossifications (61%) (Figure 2b). Obesity and short stature were reported in 53% and 44% of the cases, respectively.

A limited number of studies reported adjunctive endocrinological information such as TSH, PTH, GHRH, LH, and FSH resistance. TSH resistance was described in the totality of the cases (20/20) whereas PTH, GHRH, and LH/FSH resistance were reported in 90%, 69%, and 6% of the cases, respectively.

Other clinical features, such as seizures, brain calcification, and growth restriction, were overall poorly investigated by the included studies. However, in these selected studies the prevalence of seizures reached a percentage close to 40%.

## 4. Discussion

In the current article we reported two cases of PHP1a associated with the c.565_568delGACT variant in the *GNAS* gene and performed a literature review searching for the previously described phenotypes associated with this mutation hotspot.

A total of 76 patients were collected. The most frequent diagnosis associated with the c.565_568delGACT variant was PHP1a (around 80% of the cases). The prevalence of clinical characteristics in patients with PHP1a is reported in Figure 1. The most frequent clinical features in patients with c.565_568delGACT-related PHP1a were brachydactyly, round face, intellectual disability, and subcutaneous/heterotopic ossifications (Table 1).

To investigate whether these frequencies were in line with the current literature on PHP1a, we referred to the most recent systematic review on genotype–phenotype correlation in PHP1a [8]. Jiang et al. described the incidence of the main features in patients with PHP1a depending on the different mutation types. They grouped the mutation types into four categories; two of them included missense and in-frame mutations whereas the third and the fourth groups included patients with loss-of-function variants. In the third group that included loss-of-function variants with strong or very strong evidence of pathogenicity, part of the casuistry was composed by the hotspot deletion in exon 7. Although they confirmed the highest frequency of the c.565_568delGACT mutation in the population, they did not provide a separate incidence of the clinical characteristics for this variant. So, we tried to fill this gap through the present work.

For this reason, we compared the prevalence of signs and symptoms in PHP1a associated with the c.565_568delGACT variant with the clinical features of both all the PHPa1 (regardless of the mutation type) and the PHP1a associated with loss-of-function variants (Table 1).

In PHP1a associated with the c.565_568delGACT mutation, a lower incidence of endocrinological abnormalities at diagnosis, as compared to the other two categories (Table 1), was evident. It is interesting that the category of the loss-of-function *GNAS* variants (to which our mutation belongs) also showed a higher incidence of hypocalcemia and hyperphosphatemia [8]. A possible explanation for this difference could be that the presence of other clinical characteristic features in c.565_568delGACT-related PHP1a could have favored the diagnosis in absence of endocrinological defects.

The higher sensitivity of PTH than other laboratory findings (hypocalcemia and/or hyperphosphatemia) is in line with the previous findings and with the current theories hypothesizing that PTH increases in the earlier stage of the disease in trying to compensate for the subclinical blood calcium deficit. However, subsequently, this mechanism no longer works leading to blood calcium and phosphate changes.

AHO features showed an overall high prevalence in PHP1a (Table 1). Of them, a higher prevalence of brachydactyly, round face, short stature, intellectual disability, and subcutaneous/heterotopic ossifications was evident in PHP1a with the c.565_568delGACT variant.

Although, some of them, such as brachydactyly and round face, that could be useful in facilitating the diagnostic suspect of a PHP1a, have a restricted clinical resonance. On the contrary, features such as heterotopic ossifications, short stature, and intellectual disability have an influence on some therapeutic choices.

Subcutaneous and/or heterotopic ossifications (HO) significantly affect the quality of life of the patients, especially in consideration of the non-infrequent nerve impingement with subsequent pain, contractures, impaired movements, and pressure ulcers. HO benefits from both prophylactic and therapeutic approaches [15]. The prophylactic strategy aims at preventing or reducing the extent of HO while the pharmacological/surgical ones are mainly symptomatic and improve the HO-related disturbances once the condition has occurred. Prophylactic strategies include low-dose radiation, nonsteroidal anti-inflammatory drugs (e.g., indomethacin), bisphosphonate, and novel pharmacological compound, such as retinoic acid receptor (RARγ) agonists and free-radical scavengers [16]. Physical therapy includes Range of Motion (ROM) exercises that help prevent muscle atrophy and preserve joint motion. Additionally, they can benefit from a surgical treatment that consists of the resection of mature bone only once the HO has fully matured (which commonly happens from 12 to 18 months after the initial presentation) [15].

Short stature is another aspect that is gaining a growing interest in the genetic landscape due to the promising responses of several genetic syndromes (e.g., Noonan, Prader-Willi, Turner, Down, etc.) to growth hormone (GH) replacement therapies [17,18,19,20]. Recently, Ertl et al. conducted a multicentric study on *GNAS*-related short stature, showing a gain in height of 0.7 standard deviation scores (SDS) after 1 year (and of 1.5 SDS after 3 years), as well as the clear beneficial impact of RHGH on adult height when compared to untreated controls, with a significant difference of 1.9 SDS [21]. So, knowing that almost half of the patients with PHP1a associated with the c.565_568delGACT variant manifest short stature lets us plan specific treatments with GH replacement therapy.

Similarly, knowing that more than two-thirds of the PHP1a associated with the c.565_568delGACT variant have an intellectual disability is a useful weapon to fight the difficulties in the everyday lives of these patients. In fact, the management of intellectual disabilities must begin promptly to prevent their further worsening, to minimize the symptoms of the disability, and to improve the quality of the everyday lives of those affected. The educational support teaches children how to seek assistance, as well as behavioral, communication, and functional living and social skills on the basis of individual needs in the least restrictive environment. Behavioral therapy supports the correct interaction of the patients with the surrounding environment by encouraging positive behaviors and discouraging undesirable behaviors. Psychopharmacologic treatments are useful instruments to handle aggressive behaviors, ADHD, depression, and movement disorders that represent common comorbid conditions accompanying intellectual disability. Another important aspect is the family education addressed to the family members of these patients. It is crucial to offer specific caregiver training and also to recognize family members that bear an excessive amount of stress early. The family support must include psychosocial evaluations, psychotherapeutic approaches, and pharmacological treatments to handle depression and anxiety.

It is interesting that some of these features are shared by other genetic disorders, most of which show a defect in G protein signaling. In fact, the important role played by the Gsα protein and, in general, by G protein signaling (especially in the central nervous system) explains the clinical overlap with other clinical conditions, including specific genetic disorders and neurological diseases in which a dysfunction in this pathway has been recently reported [22]. An example could be represented by Wolfram syndrome, which shares some aspects with *GNAS*-related disorders, in particular, neurological involvement and the growth hormone deficiency [23], and in which G protein signaling deficits have been reported, suggesting a possible role in determining part of its clinical features [24,25]. Overall, G protein-coupled receptors play an important role for the central nervous system in the response to the external stimuli, and this explains why the G protein signaling pathways have a crucial role in several neurological disorders including Alzheimer’s disease, Huntington’s disease, Parkinson’s disease, and multiple sclerosis, as well as in psychiatric illnesses such as schizophrenia, bipolar disorder, depression, and attention deficit hyperactivity disorder [26,27,28]. For this reason, to date, the G protein pathway represents a promising potential target for all these conditions [28].

In summary, in the present article, we provide a genotype-phenotype correlation of PHP1a disorders associated with the c.565_568delGACT variant. This improves the surveillance of the disease since it provides information about the signs and symptoms that could be associated with the specific mutation, providing the opportunity to handle the various aspects at best (Figure 3). The genotype-phenotype correlation has an important large-scale impact, considering that the current mutation is to date considered the most common in the population.

The main limit of this study is probably represented by the reference data of the patients with genetically determined PHP1a. These data refer to the current literature and especially to the most recent review conducted by Jang et al. where the authors reported the total prevalence of some clinical features in the PHP1a associated with a *GNAS* mutation [8]. These data represent the prevalence, regardless of the mutation type and so also contain data about the c.565_568delGACT mutation. This contributes to making less evident the differences between the specific presentation associated with the c.565_568delGACT variant and the other PHP1a associated with a *GNAS* mutation.

## Figures and Tables

**Figure 1 ijms-25-10913-f001:**
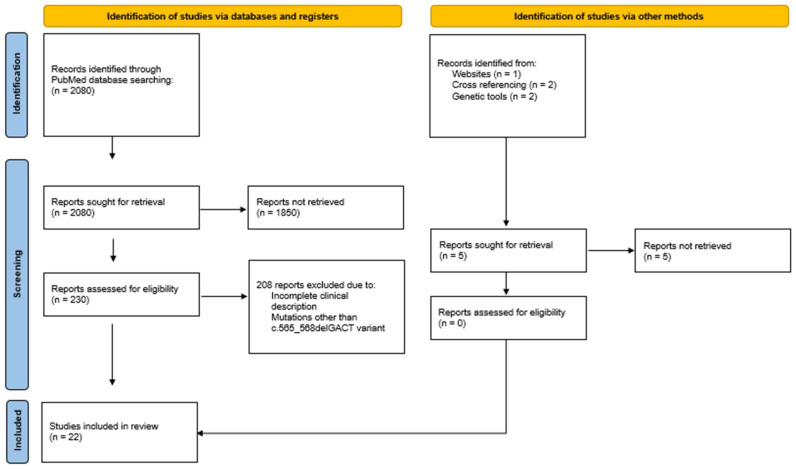
Search strategy and selection.

**Figure 2 ijms-25-10913-f002:**
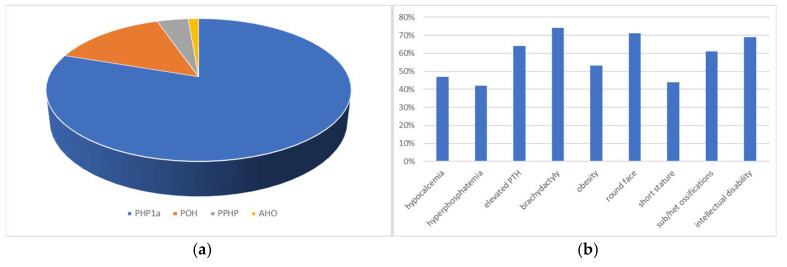
Prevalence of c.565_568delGACT-related disorders and clinical features associated with PHP1a due to c.565_568delGACT variant. Sub/het ossifications = subcutaneous/heterotopic ossifications.

**Figure 3 ijms-25-10913-f003:**
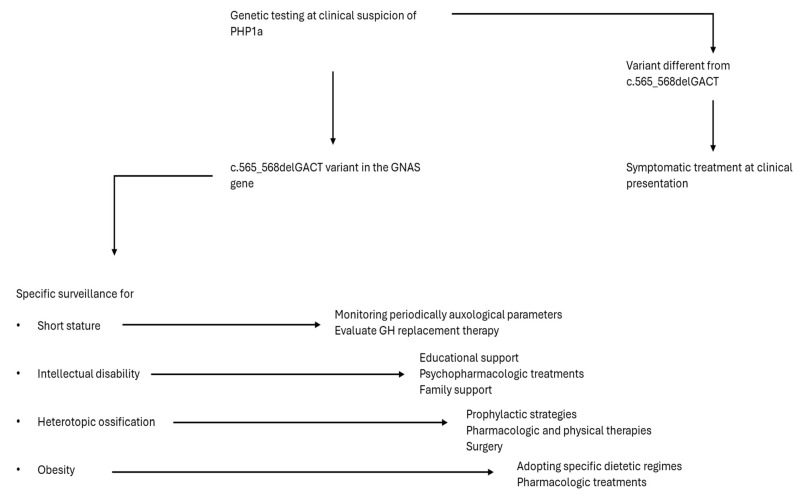
Specific surveillance in PHP1a disorders associated with the c.565_568delGACT variant.

**Table 1 ijms-25-10913-t001:** Prevalence of clinical characteristics in PHP1a according to the mutation type.

Clinical Features	Prevalence Regardless of the Mutation Type (According to [8]	Prevalence in LOF Variants with PVS1 or PVS1_Strong Strength Levels [8]	Prevalence in the *GNAS* c.565_568delGACT Variant
Hypocalcemia	54.3%	57.8%	47%
Hyperphosphatemia	60.1%	63.0%	42%
PTH increased	80.0%	85.9%	64%
TSH resistance	75.5%	75.8%	100%
GHRH resistance	18.0%	19.0%	69%
LH/FSH resistance	6.3%	6.1%	6%
AHO clinical features	87.5%	88.7%	
Brachydactyly	59.2%	58.1%	74%
Short stature	25%	24.2%	44%
Intellectual disability	28.8%	28.7%	69%
Round face	33.2%	39.1%	71%
Obesity	52.8%	55.7%	53%
Subcutaneous ossifications	39.1%	52.3%	61%

## Data Availability

The data that support the findings of this study are available from the corresponding author, C.P., upon reasonable request.

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
