# Peer review of "Genotype–Phenotype Correlation of GNAS Gene: Review and Disease Management of a Hotspot Mutation"

_ijms, 2024, doi:10.3390/ijms252010913_

Round 1

Reviewer 1 Report

Comments and Suggestions for Authors

There is a formal aspect not satisfied by this review: according to the indications of IJMS (similarly to most journals) novel data cannot be included in a review . This article (classified as Review) is a hybrid as the Authors describe two novel PHP 1a cases clinically and genetically characterized in their institutions  both harboring the most common mutation detected so far in PHP 1a patients. Section 3 - Results (as in a research article) includes  the detailed description  of Patient 1 (3.2.1) and Patient 2 (3.2.2). Then the Authors extend  to December 15 2023 the review of the literature on genotype- phenotype correlations in Pseudohypoparathyroidism type 1a  (PHP 1a) patients made by Yang till May 31 2021 (ref 8), but  instead of considering all the causative pathogenic variants focus on the patients with  the most frequent mutation detected so far (GNAS exon 7 c.565_568delGACT).  This choice makes the clinical signs associated with this mutation more incisive raising the challenge of personalized follow-up and management of the carrier patients.

The Authors should restructure their manuscript  as a research  article which describes their new cases with review of the literature .

As regards the content I have the following criticisms:

Replace “mental retardation” with “intellectual disability” throughout the text and in the Figures

The c.565_568delGACT here at focus  is named  c.2494 2497del-GACT when the Authors describe  the results of WES in their patients 1 and 2. Of course a geneticist knows that it is the same mutation with a different mRNA transcript reference, though a clinical geneticist reader may not understand. Please homogenize or give the appropriate coordinates (NM 000516.4  and NM 080425.2), respectively.

It would be interesting to analyze the sequence of GNAS exon 7 for evolutionary conservation, GC content. SINE and LINE elements, tandem repeats, which might concur to the instability and proneness of exon 7 to this most common pathogenic variant.

Minor criticism

Page 2 : According to this more recent classification……….as iPPSD2 that includes the previously called PHP type 1a and 1a (1b?)

Author Response

Comments 1:

There is a formal aspect not satisfied by this review: according to the indications of IJMS (similarly to most journals) novel data cannot be included in a review . This article (classified as Review) is a hybrid as the Authors describe two novel PHP 1a cases clinically and genetically characterized in their institutions  both harboring the most common mutation detected so far in PHP 1a patients. Section 3 - Results (as in a research article) includes  the detailed description  of Patient 1 (3.2.1) and Patient 2 (3.2.2). Then the Authors extend  to December 15 2023 the review of the literature on genotype- phenotype correlations in Pseudohypoparathyroidism type 1a  (PHP 1a) patients made by Yang till May 31 2021 (ref 8), but  instead of considering all the causative pathogenic variants focus on the patients with  the most frequent mutation detected so far (GNAS exon 7 c.565_568delGACT).  This choice makes the clinical signs associated with this mutation more incisive raising the challenge of personalized follow-up and management of the carrier patients.

The Authors should restructure their manuscript  as a research  article which describes their new cases with review of the literature .

Response 1:

We thank the reviewer for the comment and apologize for this oversight. We have revised the article by removing the term 'systematic' and reorganized the methods, omitting certain aspects (such as the risk of bias) that are important for systematic reviews but not for other types of reviews. We have kept the flow chart as it is intuitive and still allows for the data extraction process to be followed by the reader.

Comments 2:

As regards the content I have the following criticisms:

Replace “mental retardation” with “intellectual disability” throughout the text and in the Figures

Response 2:

We replaced it.

Comments 3:

The c.565_568delGACT here at focus  is named  c.2494 2497del-GACT when the Authors describe  the results of WES in their patients 1 and 2. Of course a geneticist knows that it is the same mutation with a different mRNA transcript reference, though a clinical geneticist reader may not understand. Please homogenize or give the appropriate coordinates (NM 000516.4  and NM 080425.2), respectively.

Response 3:

We thank the reviewer for the comment that improved the readability and the accessibility to the manuscript; we have standardized the terminology, providing a consistent naming of the variant to avoid any misunderstandings.

Comments 4:

It would be interesting to analyze the sequence of GNAS exon 7 for evolutionary conservation, GC content. SINE and LINE elements, tandem repeats, which might concur to the instability and proneness of exon 7 to this most common pathogenic variant.

Response 4:

We thank the reviewer for the insightful suggestion. We agree that analyzing the evolutionary conservation, GC content, and presence of SINE/LINE elements and tandem repeats in GNAS exon 7 could provide valuable insights into its instability and susceptibility to pathogenic variants. While these analyses fall outside the scope of the current study, they represent interesting avenues for future research and we do not exclude the possibility of addressing them in future work.

Comments 5:

Minor criticism

Page 2 : According to this more recent classification……….as iPPSD2 that includes the previously called PHP type 1a and 1a (1b?)

Response 5:

We apologize for the error, we modified it.

Reviewer 2 Report

Comments and Suggestions for Authors

This is a very interesting paper on a very complex topic, the GNAS gene. This locus has a highly complex imprinted expression pattern. It gives rise to maternally, paternally, and biallelically expressed transcripts that are derived from four alternative promoters and 5' exons. Some transcripts contain a differentially methylated region (DMR) at their 5' exons, and this DMR is commonly found in imprinted genes and correlates with transcript expression. An antisense transcript is produced from an overlapping locus on the opposite strand. So, this is a locus with many different genetic products, and Gsalpha is commonly considered the most relevant.

I found it interesting that the GNAS deficiency has diverse clinical features. This reminds us of Wolfram syndrome, which has many similar features (growth hormone deficiency, neurodegeneration), albeit not all.

1. Gsalpha deficiency or deficiency in G-protein regulation seems to be a feature of Wolfram syndrome as well, and the authors should discuss this, considering these papers (PMID: 19293327, 37759745, 22028430).

2. Gsalpha and G-protein regulating genes are also involved in polygenic conditions like Parkinson's disease (PMID: 38740892). this is also something that the authors can discuss. That would improve the translational potential of the paper.

Author Response

Comments 1:

This is a very interesting paper on a very complex topic, the GNAS gene. This locus has a highly complex imprinted expression pattern. It gives rise to maternally, paternally, and biallelically expressed transcripts that are derived from four alternative promoters and 5' exons. Some transcripts contain a differentially methylated region (DMR) at their 5' exons, and this DMR is commonly found in imprinted genes and correlates with transcript expression. An antisense transcript is produced from an overlapping locus on the opposite strand. So, this is a locus with many different genetic products, and Gsalpha is commonly considered the most relevant.

I found it interesting that the GNAS deficiency has diverse clinical features. This reminds us of Wolfram syndrome, which has many similar features (growth hormone deficiency, neurodegeneration), albeit not all.

  1. Gsalpha deficiency or deficiency in G-protein regulation seems to be a feature of Wolfram syndrome as well, and the authors should discuss this, considering these papers (PMID: 19293327, 37759745, 22028430).

  1. Gsalpha and G-protein regulating genes are also involved in polygenic conditions like Parkinson's disease (PMID: 38740892). this is also something that the authors can discuss. That would improve the translational potential of the paper.

Response 1:

We thank the reviewer for both these comments. We added a paragraph on these aspects in the discussion section. In detail, “It is interesting that some of these features are shared by other genetic disorders, most of which show a defect in the G protein signaling. In fact, the important role played by the gsα protein and, in general, by the G protein signaling (especially in the central nervous system) explains the clinical overlap with other clinical conditions including specific genetic disorders and neurological diseases in which a dysfunction of this pathway has been recently reported (Schöneberg et al. 2024). An example could be represented by the Wolfram syndrome that shares some aspects with GNAS related disorders, in particular the neurological involvement and the growth hormone deficiency (Kõks 2023), and in which G protein signaling deficits have been reported, suggesting a possible role in determining part of its clinical features (Kõks et al. 2009; Kõks et al. 2011). Overall, G protein-coupled receptors play an important role for the central nervous system in the response to the external stimuli and this explains why the G protein signaling pathways have a crucial role in several neurological disorders including Alzheimer’s disease, Huntington’s disease, Parkinson’s disease and Multiple sclerosis as well as in psychiatric illnesses such as schizophrenia, bipolar disorder, depression, Attention deficit hyperactivity disorder (Kalinovic et al. 2024; Fröhlich et al. 2024; Azam et al. 2020). For this reason, to date, the G protein pathway represents a promising potential target for all these conditions (Azam et al. 2020).”

Round 2

Reviewer 1 Report

Comments and Suggestions for Authors

The manuscript has been reorganised achieving a better and more punctual format. The nomenclature of the GNAS mutation at focus has been unified in the carrier literature' patients and the new ones characterized by the Authors.  The take home message is clear to the reader. The comment  on shared similarities between most GNAS patient with the common variant and patients with defects in  the G-protein signaling, add to the  the  discussion. 

Reviewer 2 Report

Comments and Suggestions for Authors

All my comments were addressed. The manuscript is ready to be accepted.